# Mucosal Vaccination with *Lactococcus lactis*-Secreting Surface Immunological Protein Induces Humoral and Cellular Immune Protection against Group B *Streptococcus* in a Murine Model

**DOI:** 10.3390/vaccines8020146

**Published:** 2020-03-26

**Authors:** Diego A. Diaz-Dinamarca, Carlos Hernandez, Daniel F. Escobar, Daniel A. Soto, Guillermo A. Muñoz, Jesús F. Badilla, Ricardo A. Manzo, Flavio Carrión, Alexis M. Kalergis, Abel E. Vasquez

**Affiliations:** 1Sección de Biotecnología, Instituto de Salud Pública de Chile, Santiago 780050, Chile; dadiaz@ispch.cl (D.A.D.-D.); carlos.hernandez.a@ug.uchile.cl (C.H.); descobar@ispch.cl (D.F.E.); dasoto@ispch.cl (D.A.S.); guillermomunozbriones@gmail.com (G.A.M.); jesusbadillaya@gmail.com (J.F.B.); rmanzo@ispch.cl (R.A.M.); 2Millennium Institute of Immunology and Immunotherapy, Departamento de Genética Molecular y Microbiología, Facultad de Ciencias Biológicas, Pontificia Universidad Católica de Chile, Santiago 8380453, Chile; akalergis@bio.puc.cl; 3Departamento de Bioquímica y Biología Molecular, Facultad de Ciencias Químicas y Farmaceuticas, Universidad de Chile, Independencia, Santiago 8380492, Chile; 4Escuela de Biotecnología, Facultad de Ciencias, Universidad Santo Tomas, Santiago 8320000, Chile; 5Programa de Inmunología Traslacional, Facultad de Medicina, Clínica Alemana Universidad del Desarrollo, Santiago 7610315, Chile; fcarrion@udd.cl; 6Departamento de Endocrinología, Facultad de Medicina, Pontificia Universidad Católica de Chile, Santiago 8330077, Chile; 7Facultad de Medicina y Ciencia, Universidad San Sebastián, Providencia, Santiago 8320000, Chile

**Keywords:** group B streptococcus, mucosal vaccine, surface immunogenic protein, cellular immune response, humoral immune response

## Abstract

Group B *Streptococcus* (GBS) is the primary etiological agent of sepsis and meningitis in newborns and is associated with premature birth and stillbirth. The development of a licensed vaccine is one of the pending challenges for the World Health Organization. Previously, we showed that oral immunization with surface immune protein (SIP) decreases vaginal colonization of GBS and generates functional opsonizing antibodies, which was determined by opsonophagocytic assays (OPA) in vitro. We also showed that the protein has an adjuvant vaccine profile. Therefore, an oral vaccine based on SIP may be an attractive alternative to employ in the development of new vaccines against GBS. *Lactococcus lactis* is a highlighted oral vaccine probiotic inducer of the mucosal immune response. This bacterium could serve as an antigen-delivering vehicle for the development of an edible vaccine and has been used in clinical trials. In this study, we showed that an oral vaccine with a recombinant *L. lactis* strain secreting SIP from GBS (*rL. lactis*-SIP) can induce protective humoral and cellular immunity in an experimental model of GBS vaginal colonization in C57BL/6 mice. Mice immunized with *rL. lactis-*SIP were protected against clinical symptoms and bacterial colonization after GBS vaginal colonization. Our *rL. lactis-*SIP vaccine also induces an increase of immunoglobulin G (IgG) and immunoglobulin A (IgA) specifically against SIP. The adoptive transfer of serum from vaccinated mice to naïve mice generated protection against GBS vaginal colonization. Moreover, the *rL.*
*lactis*-SIP strain induces the activation of SIP-specific T cells, which could decrease GBS vaginal colonization and generate protective antibodies when transferred to other mice. Our experimental observations strongly support the notion that *rL. lactis*-SIP induces protective humoral and cellular immunity and could be considered as a novel alternative in the development of vaccines for GBS.

## 1. Introduction

Group B *Streptococcus* (GBS), also known as *Streptococcus agalactiae*, is a Gram-positive, facultative anaerobe and opportunist pathogen that is β-hemolytic on lamb blood agar. GBS colonizes the gastrointestinal and genitourinary tracts of more than 30% of the healthy population. The global prevalence of maternal rectovaginal colonization is reported to be in the range of 11–35% [1,2,3]. In Chile, the prevalence was determined as being in the range of 11–20% [4]. GBS infection is responsible for 90,000 deaths among infants <3 months age and 57,000 fetal infections/stillbirths worldwide [5]. To date, there are no licensed vaccines for GBS, and phase I and II clinical trials are under way with hexavalent conjugate vaccines and protein-based vaccines [6,7]. Serotype-specific multivalent vaccines are being developed for delivery to pregnant women to protect their infants from GBS [6]. Polysaccharide-based vaccines induce an efficient protective immune response, such as partial coverage and induced escape, leading to persistence of disease [7]. Furthermore, GBS has been reported to have a capsular Switching from capsular polysaccharide (CPS) [8]. A vaccine based on a conserved and immunogenic protein could be an interesting alternative for the development of a vaccine against GBS [8]. A maternal vaccine given to pregnant women to stimulate the passive transplacental transfer of protective antibodies has the potential to reduce maternal disease, adverse pregnancy outcomes, and newborn disease [6,9].

GBS has evolved a series of innate defense evasion molecules that can evade complement proteolytic cascade, which allows them to overcome immune clearance by the host [10]. Moreover, GBS can bind sialic acid-binding immunoglobulin-like lectins (Siglecs) through the sialic acid capsule or b-protein to suppress immune cell activation and placental membrane inflammation [11], which could potentially lead to increased rates of GBS-associated preterm birth and stillbirth. Thus, vaccine-mediated immunity against GBS could be elicited by the combination of strong humoral and cell-mediated immune responses. Surface immunogenic protein (SIP) is a highly conserved protein that is present among all GBS serotypes (Ia, Ib, II-IX) [12,13]. An animal model demonstrated that specific antibodies against SIP could cross the placenta to confer protection to newborns against diseases caused by GBS [14]. We also recently reported that SIP is a toll-like receptor (TLR)-2 and TLR-4 protein agonist, suggesting potential as a new immune adjuvant [15]. In addition, SIP has been used in the development of a vaccine for GBS infection in a Nile tilapia (*Oreochromis niloticus*) model and generated an increase in innate immunity [16,17]. It has been shown that subcutaneous and oral immunization with SIP induces a protective humoral and cellular immune response [18,19]. In the context of oral immunization strategies, the route of administration would be adequate for the generation of mucosal immunity, which has already been proven to generate protection against diseases such as cholera and polio [20,21]. An oral vaccine might be advantageous for the distribution and administration of GBS vaccines, which could improve the uptake of antigen vaccination and overall public health.

*Lactococcus lactis* is a Gram-positive organism that is non-pathogenic and non-invasive that belongs to the group of lactic bacteria. Furthermore, this organism is “generally regarded as safe” (GRAS) [22]. Viral, bacterial, and parasitic antigens have been expressed in *L. lactis*, and recombinant strains have been shown to be capable of delivering antigens at the intestinal mucosal sites, inducing a specific immune response in mice [23,24]. It has also been demonstrated that *L. lactis* can survive passage through the gastrointestinal tract in animals and human volunteers without colonizing it, which makes it a good candidate for vaccines in humans [25,26,27]. Immunization with *L. lactis* has advantages due to the vaccine adjuvant properties of its peptidoglycan wall and can also be used as a protein expression system that ensures the expression of the antigen [28]. In this context, the expression of SIP as an antigen in an *L. lactis* strain could serve as an antigen-delivering vehicle for the development of an edible vaccine. In this study, we evaluated whether *L. lactis* strains that secrete SIP from GBS could promote a protective humoral and cellular immune response against GBS colonization in a mouse model.

Our data showed that vaccination with a recombinant *L. lactis* strain secreting SIP (*rL. lactis*-SIP) efficiently induced protective immunity against bacterial colonization. Furthermore, this vaccine was able to stimulate lymphoid tissue associated with intestine dendritic cells and increase mucosal immunoglobulin A (IgA) against SIP. We also observed that the protection was mediated by the humoral and T cell response, which was able to prevent GBS vaginal colonization when transferred to other mice. Moreover, our vaccine stimulates CD4 T lymphocytes (CD4 + T cells) that generate opsonophagocytic functional antibodies in naïve mice. These results suggest that *rL. lactis*-SIP is an efficient prophylactic strategy that can induce a cellular immune response capable of boosting the humoral immune response against SIP and generate protective immunity against GBS infection.

## 2. Materials and Methods

### 2.1. Ethics Statement

All experiments that used mice were conducted in accordance with the international ethical standards and Chilean Law 20380 on Animal Protection (2009). The experimental protocol was reviewed and approved by the Institutional Committee of Care and Use of Laboratory Animals of Public Health Institute of Chile (approval no. C030519-5).

### 2.2. Mouse Strains

Six-to-eight-week-old C57BL/6 wild-type (WT) female mice were purchased from Jackson Laboratories (Bar Harbor, ME, USA) and maintained in a pathogen-free animal facility at the Public Health Institute of Chile. 

### 2.3. Production of Recombinant L. lactis that Secretes rSIP from GBS

The plasmid vector pNZ8124::*sip* was constructed by GenScript (New Jersey, United States, Project: U1871DC290-3 pNZ8124). Briefly, the *sip* gene (GeneBank accession no. KX363665.1) was truncated in its secretion sequence for a GBS signal (N-terminal of SIP containing a 25-amino-acid signal peptide that is cleaved in the mature protein from GBS) and were chemically synthesized with EcoR V (recognition site GAT^ATC ) and Sal I (G^TCGAC) restriction sites at the 5′ and 3′ ends. The *sip* gene was cloned into the expression plasmid pNZ8124 (MoBiTec, Göttingen, Germany). The cloned vector pNZ8124::*sip* contains the sequence of the lactococcal signal peptide Usp45 (SP usp45) [29] fused to the PnisA promoter, which is inducible by nisin, thereby resulting in pNZ8124::*sip*. The *Escherichia coli* (*E. coli)* strain MC1061 was used for the propagation of the expression vector. 

The construct pNZ8124::*sip* was transformed into electrocompetent *L. lactis* NZ9000 strain, resulting in r*L. lactis*-SIP. The colonies that could grow on brain heart infusion (BHI) chloramphenicol agar (10 μg/mL) were selected. A colony obtained from the transformation was inoculated in 5 mL of BHI medium supplemented with chloramphenicol (10 μg/mL) and incubated at 30 °C overnight. The presence of vector pNZ8124::*sip* was verified by PCR. Recombinant *L. lactis* was used to characterize the expression of SIP (*rL. lactis-*SIP) by SDS-PAGE and Western blot analyses.

### 2.4. Secretion of SIP from GBS in L. lactis NZ9000

A bacterial inoculum of 5 mL was grown overnight in a BHI medium containing 10 μg/mL of chloramphenicol at 30 °C without agitation. Next, 400 μL of the previous inoculum was placed in a final volume of 10 mL of BHI medium, which was incubated at 30 °C without agitation until an OD_600nm_ of 0.4 was reached and induced with 10 ng/mL of nisin for 3 h at 30 °C without agitation. The medium was centrifuged at 5000× *g* for 10 min at 4 °C, and then the supernatant was removed and concentrated using Amicon Ultra-15 10K centrifugal filters. Halt 100× protease inhibitor was added (1×), and the medium was stored at −20 °C until further characterization analysis. 

For oral immunization, the strain of *L. lactis* NZ9000 (*L. lactis*) and recombinant *L. lactis* NZ9000 secreting SIP (*rL. lactis*-SIP) were cultured as described above, and induced for 1 h with nisin. A volume of 50 mL of cell culture was collected by centrifugation at 13,000*× g* for 10 min and resuspended in 500 μL of sterile phosphate-buffered saline (PBS) 1×. A final concentration of ≈1 × 10^11^ colony forming units (CFU)/mL was obtained for immunization by oral administration.

### 2.5. Oral Immunization and GBS Vaginal Colonization

The immunization model was modified from Diaz-Dinamarca et al. [30]. C57BL/6 female mice at 6 to 8 weeks of age (obtained from the Institute of Public Health of Chile) were acclimated and randomized into experimental groups. The animals were kept in conditions free of standard pathogens and were given free access to food and water during the experiment. The mice were manipulated and eliminated according to the guidelines of the Institutional Ethics Committee.

We developed two immunization schemes. For the evaluation of immune response, we performed four immunizations (on days 1, 14, 28, and 42). This scheme had an extra booster compared to the scheme described below, which was used in order to enable the generation of a supposed “protective” level of effector T cells [31]. This model ended on day 43, when the animals were euthanized to obtain the genitouterine tract, blood, spleen, and small intestine. For the second immunization scheme, we used three immunizations on days 1, 14, and 28 and evaluated the decrease in colonization by GBS.

GBS vaginal colonization was performed according to the report by Soto et al. [19]. Briefly, from days 29 to 34, the animals received a daily dose of 100 mg/kg of gentamicin to decrease the vaginal commensal flora, as well as 0.1 mg of 17β-estradiol to synchronize the estral cycle in estrus. On day 35, 1 × 10^7^ CFU of GBS in gelatin (10%) was inoculated into the genitourinary tracts of all animals. The model ended on day 41, when the animals were anesthetized and sacrificed by cervical dislocation, and samples of vaginal lavage and blood were obtained.

### 2.6. Opsonophagocytic Assay

Human promyelocytic leukemia cells (HL-60 cells; CCL-240; ATCC) were propagated in Roswell park memorial institute (RPMI) 1640 containing 10% fetal bovine serum (FBS) supplemented with 1% L-glutamine, 100 U/mL penicillin, and 100 g/mL streptomycin. The cells were differentiated into neutrophils in RPMI medium without antibiotics but containing 10% fetal calf serum, glutamine, and 1.3% dimethyl sulfoxide (DMSO) for 6 to 7 days [32]. HL-60 differentiation was about 95% of the cells. Characterization was performed by flow cytometry using CD11b expression (data not shown).

The opsonophagocytosis assay (OPA) was conducted on serum from immunized mice, as described by Guttormsen et al. [32]. A hemolytic GBS strain (Genbank Code: KU736792) was grown in sealed tubes containing Todd–Hewitt broth. The GBS was washed with PBS, resuspended in modified Eagle’s medium, and used in the assay. Briefly, the reaction was performed in 96-well plates (Nunc) in Hank’s balanced salt solution (HBSS, Gibco). For each reaction mixture, heat-inactivated test serum (56 °C for 30 min; HI), GBS bacteria, differentiated HL-60 cells, and 10% baby rabbit complement (Cedarlane) were added.

Control reactions were carried out without complement or antibody, effector cells, or all components except GBS. The ratio of effector cells to GBS cells was 90:1. The reaction mixtures were incubated at 37 °C for 1 h with shaking. Aliquots were removed after incubation and plated on blood agar plates, which were incubated overnight at 37 °C in 5% CO_2_. The percentage killed was assessed as described by Romero-Saavedra et al. [33] by comparing the colony counts at 60 min (t60) that did not contain differentiated HL-60 cells to the colony counts of a tube that included all four components of the assay.

### 2.7. Analysis of Immunoglobulins from Small Intestine and Feces

Immunoglobulins were obtained from the small intestine by placing the organ in a Petri dish and washing it with 4 mL of Ca^2+^/Mg^2+^-free PBS plus Halt 100× protease inhibitor (1×). The intestine contents were vortexed and then centrifuged at 5000× *g* for 10 min at 4 °C. Stool was collected by changing the cage of the mice half an hour before sacrificing them. The harvested feces were weighed and diluted in 1× PBS plus Halt 100× protease inhibitor (1×) to a final concentration of 0.2 mg/mL. The supernatants of feces and intestines were frozen at −80 °C for the quantification of IgA by ELISA.

### 2.8. Measures of Immunoglobulins by ELISA

Serum immunoglobulin G IgG and IgA from intestinal washings were detected according to procedure described by Diaz-Dinamarca et al. [30]. Briefly, a 96-well plate was activated with 1 μg of recombinant SIP (rSIP). The plate was then washed three times with PBS-0.05% Tween (PBST) and blocked for 1 h at 37 °C with PBS-5% skimmed milk. After washing the plate, sample was added to each well in triplicate with serial dilutions of 1:10 to 1:1000 in PBS-1% skimmed milk and incubated for 1 h at 37 °C. After washing, anti-IgG and mouse IgA antibodies conjugated to HRP were added at a 1:5000 dilution in PBS-1% skimmed milk. After 1 h, the sample was revealed by the reagent one-step ultra 3,3’,5,5’-tetramethylbenzidine (TMB)-substrate (Thermo). The reaction was stopped by adding 2M sulfuric acid to each well, and the absorbance was read at 450 nm using a Spectro Star Nano ELISA plate reader (BMG Labtech).

### 2.9. Analysis of Small Intestinal Mucosal Cells and T Cell Activation

Small intestinal mucosal cells were isolated, and the epithelial layer cells were purified separately using 2 mM ethylenediaminetetraacetic acid (EDTA) (1.5 mg/mL collagenase type II (Gibco) in Ca^2+^- and Mg^2+^-free Hank’s balanced salt solution, respectively, as described by Hurasato et al. [34]. These cells were stained with anti- major histocompatibility complex MHC II PE (phycoerythrin) (BD Pharmingen, clone AF6-120.1), anti-CD11b FITC (fluorescein isothiocyanate) (BD Pharmingen, clone M1/70), anti-CD11c (BD Pharmingen, clone HL3), and anti-CD103 APC (allophycocyanin) (BD Pharmingen, clone M290).

The spleens were removed from all the experimental mice at day 1 after the last immunization. The spleens were perfused with PBS-1x, and then the cell suspension was centrifuged at 300× *g* and resuspended in RPMI-1640 + 10% FBS for 5 min. The pellets were washed, counted in a hematologic chamber, and stained with anti-CD45 FITC (BD Pharmingen, clone 30-F11), anti-CD4 PE (BD Pharmingen, clone RM4-5), and anti-CD69 APC (BD Pharmingen, clone H1.2F3). Data were acquired using a FACSVerse flow cytometer (BD Biosciences) and analyzed using the software FlowJo 7.6.1.

### 2.10. Analysis of Regulatory T cells (Tregs)

Spleen cells were measured after 5 days of stimulation. The proportions of CD4+CD25+Foxp3+ (forkhead box P3) Tregs in the cell mixtures were analyzed by flow cytometry using a True-Nuclear One-Step Staining Mouse Treg Flow kit (FOXP3 Alexa Fluor 488/CD25 PE/CD4 PerCP; Biolegend). The results are expressed as the percentages of CD4+CD25+Foxp3+T cells in CD4+ cells.

### 2.11. Relative Quantification of Transcriptions Factors by Real-Time PCR

Following the manufacturer’s instructions, we used the automated NucliSENS-EasyMAG platform (Biomerieux, Marcy l’Etoile, France) to purify total nucleic acids from 600,000 splenocytes contained in 500 μL of RPMI 1640 medium with 10% FBS, 25 mM β-mercaptoethanol, and penicillin/streptomycin. The splenocytes were obtained from the spleens of experimental mice and pulsed with SIP protein (purified by HPLC, as described by Diaz-Dinamarca et al. [15]). The eluates were treated with DNase (Invitrogen, Carlsbad, CA, USA) to degrade the DNA in the sample. The resulting RNA was maintained at −20 °C until complementary DNA (cDNA) synthesis. The mixture for cDNA synthesis was developed by following the manufacturer’s instructions (Agilent Technologies, Santa Clara, CA, USA) using the following thermal profile: 25 °C for 5 min, 42 °C for 45 min, 95 °C for 5 min, and final undefined step to 4 °C.

Transcription factors were analyzed using the following set of specific primers for each gene: β-actin forward 5’-AGCTGCGTTTTACACCCTTT-3´, β-actin reverse 5´-AAGCCATGCCAATGTTGTCT-3´ [35]; FOXP3 forward 5´- TTTCACCTATGCCACCCTTATC-3´, FOXP3 reverse 5´-GTAGGCGAACATGCGAGTAA-3´; signal transducer and activator of transcription (STAT)-5a forward 5´- CCGAAACCTCTGGAATCTGAA-3´ STAT5a reverse 5´-GGTCTGGGAACACGTAGATAAG-3´; STAT3 forward 5´-CCCATATCGTCTGAAACTCCTAAC-3´, STAT3 reverse 5´-TCACCCACACTCACTCATTTC - 3´. All real-time PCR amplifications were carried out in 20 μL reaction mixtures containing 2 μL of cDNA template, 10 μL of Brilliant II SYBR Green qPCR 2X (Agilent Technologies, Santa Clara, CA, USA), 1 μL of each primer (0.5 µM final concentration), and 6 μL of deionized water.

All reactions were run in quadruplicate using a Stratagene Mx3000P thermocycler (Agilent Technologies, Santa Clara, CA, USA). The following thermal profile was used in each case: initial denaturation at 95 °C for 10 min, 40 cycles of 95 °C for 30 s, and 60 °C for 1 min. Before each PCR run, the software MxPro was set up with the relative quantity option using the group of mice immunized with PBS as a calibrator and the β-actin gene (Actb) as a reference.

### 2.12. Restimulation of Splenocytes

Splenocytes were isolated from spleen, as previously described [36]. Briefly, spleens were mashed through a cell strainer (Falcon), and then the cells were spun down and resuspended in BD Pharm Lyse lysing solution. After 5 min, the cells were rinsed with complete RPMI medium containing 10% heat-inactivated FBS, 2-mercaptoethanol (50 μM), l-glutamine (2 mM), and penicillin-streptomycin (all from Gibco). Isolated splenocytes (2 × 10^6^ cells/mL) were resuspended in complete RPMI medium and restimulated with 5 μg/ml of rSIP. After 5 days, the cells were purified and transferred to recipient mice, as described below.

### 2.13. Passive Transfer of Immune IgG and T Cells (CD4+ and CD8+) from rL. lactis-Vaccinated Mice to Naïve Animals

Six-to-eight-week-old C57BL/mice were intravenously treated with 100 µL of serum from the following animal groups: a non-immunized group (PBS) and a group immunized with *rL. lactis*-SIP (*n* = 8 for each group). The serum samples used for these experiments were collected at 7 days after the last immunization. One day after the transfer, the animals were intraperitoneally anesthetized with a mixture of ketamine and xylazine (80 and 4 mg/kg, respectively) and colonized with 1 × 10^7^ CFU of GBS. The bodyweight of infected mice was recorded daily until day 7. IgG against SIP was analyzed as described above.

We also transferred activated SIP-specific T cells from immunized mice. Spleen cells from mice immunized with *rL. lactis*-SIP were cultured in the presence of rSIP obtained from *E. coli* and purified by HPLC, as described by Diaz-Dinamarca et al. [16]. Mice PBS immunized with PBS were included in these assays as a control. To confirm the T cell activation of cells specific for rSIP in spleens, cellular suspensions were stimulated with 1 µg/ml of rSIP for 4 days in vitro and then centrifuged at 300× *g* for 5 min. The pellets were then washed and stained with anti-CD45 FITC (BD Pharmingen, clone) and anti-CD25 APC (BD Pharmingen, clone PC61). Data were acquired on a FACSVerse flow cytometer (BD Biosciences) and analyzed using FlowJo 7.6.1, as described above.

After 5 days of stimulation with rSIP, spleen cells from mice immunized with *rL. lactis*-SIP and PBS were purified using MojoSort Isolation kits (Biolegend) according to the manufacturer’s instructions. A total of 1 × 10^6^ of CD4+, CD8+, or a 1:1 mixture of CD4+/CD8+ T cells was injected via the caudal vein in the recipient C57BL/6 mice. Twenty-four hours after T cell transfer, the recipient mice were colonized with 1 × 10^7^ CFU of GBS. The bodyweight and temperature of infected mice were recorded daily until day 7, and IgG against rSIP was analyzed as described above.

### 2.14. Statistical Analyses

The results are presented as the mean and standard deviation. Shapiro–Wilk test and Kolmogorov–Smirnov test were used to evaluate the normality of the distribution of the examined variables. The statistical data analysis was performed using the Student’s *t*-test and ANOVA test, and *p*-values <0.05 were considered statistically significant. The analyses were performed using GraphPad Prism software (GraphPad Software, Inc., San Diego, CA, USA)

## 3. Results

### 3.1. Secretion of rSIP by Recombinant L. lactis

The SIP from GBS was secreted using recombinant *L. lactis* under the control of the lactococcal nisA promoter. The rSIP secretion was confirmed by SDS-PAGE (Figure 1A, top) and Western blot (Figure 1A, bottom) analyses using an anti-SIP polyclonal antibody [18]. Figure 1A shows a clear band in the supernatant of *L. lactis*::pNZ8124-*sip* (*rL. lactis*-SIP) (lane 4), whereas no signal was detected in the supernatant of *L. lactis* (lane 2) and *rL. lactis-*SIP without nisin (lane 3). The positive control (lane 5) corresponded to a sample of rSIP obtained from *E. coli* BL21 (DE3) Codon Plus pET21a-*sip* and purified by a Ni-NTA (nickel-Nitrilo-Tri-Acetic-acid) affinity chromatography column. In relation to the molecular weight between SIP expressed in *E. coli* and *rL. lactis*-SIP, there is a small variation because SIP from *E. coli* contains a 6× histidine-tag and 25 amino acids of the signal peptide [12].

### 3.2. Oral Immunization with rL. lactis-SIP Decreased Vaginal-Tract Colonization by GBS in Mice

Diaz-Dinamarca et al. [18,30] showed that oral immunization with SIP and alum can reduce vaginal colonization by GBS. To evaluate whether oral immunization with *rL. lactis*-SIP decreases GBS colonization, C57BL/6 mice were immunized with 1 × 10^10^ CFU of *rL. lactis*-SIP, whereas control groups received 1 × 10^10^ CFU of *L. lactis* and 100 µL of PBS. As additional controls, mice were immunized with rSIP from *E. coli* plus alum (data not shown). At 14 and 28 days after immunization, two boosts were performed, and 5 days later, all animals were colonized with 1 × 10^7^ CFU (Figure 1B). 

We observed a high number of GBS in the vaginal tract of groups immunized with *L. lactis* and the PBS control groups. In contrast, in mice orally immunized with *rL. lactis*-SIP, we detected a significant decrease in vaginal-tract colonization by GBS (*p*-value, ** *p* < 0.01 and * *p* < 0.05) (Figure 1C). Additionally, *L. lactis* alone generated a decrease in GBS colonization compared with PBS control groups (** *p* < 0.01). The benefit of *L. lactis* as an antigen-delivering vehicle is its proteoglycan compounds in the cell wall that stimulate unspecific immune response.

We also observed clinical symptoms associated with body temperature and weight gain. Animals immunized with *rL. lactis*-SIP gained more weight compared with control groups (Figure 1D). Moreover, there was a tendency to maintain the body temperature of animals immunized with *rL. lactis*-SIP compared to control animals that had higher body temperature (data not shown). These results suggest that oral immunization with rSIP using *rL. lactis*-SIP provides a protective effect against the colonization of GBS, which was previously observed in oral immunization with recombinant SIP from *E. coli* adjuvanted with alum [30].

### 3.3. Oral Immunization with rL. Lactis-SIP Stimulated Secretions of Mucosal and Systemic Anti–SIP IgG and IgA

The use of *L. lactis* as an antigen administration vehicle as a vaccination strategy has demonstrated the ability to present antigens to the immune system and induce both humoral and cell-mediated immune responses at mucosal sites and at the systemic level. One aspect that should be considered is the effect of the immunization pathway of *L. lactis* on the immune response, given that immunogenicity has been observed as depending on the route of inoculation [22,37]. Therefore, we tested whether oral immunization with *rL. lactis-*SIP can induce a specific immune response against SIP from GBS. 

Specific immunoglobulins in serum, intestinal lavage, and feces were detected at 12 h after the last immunization using ELISA (Figure 2A). High levels of IgG response were elicited when mice were immunized with *rL. lactis*-SIP (*** *p* < 0.0001) in comparison to the IgG levels produced by the *L. lactis* and PBS control groups. In contrast, no significant difference was observed in the sera of the mice immunized with *L. lactis* in comparison to the PBS- control group (*ns*, *p* > 0.05) (Figure 2B).

We also determined the T helper (Th)1 and Th2 type response of Ig profile of immunized mice. High levels of IgG1 and IgG2a were detected in mice immunized with *rL. lactis*-SIP in comparison with the control groups. This suggests a balance in the response of the Th1/Th2 type of the group immunized with *rL. lactis*-SIP. We also evaluated the mucosal responses in vaccinated mice. For this purpose, SIP-specific IgA levels were determined in lavages of fecal samples and the small intestine (Figure 2D,E). The results indicated that vaccinated mice had higher levels of antigen-specific mucosal IgA in comparison with control mice receiving PBS or *L. lactis* (* *p* < 0.05).

### 3.4. Oral Vaccine Using rL. Lactis-SIP Increased Intestinal Antigen-Presenting Cells

The resident dendritic cells (DCs) in the intestine are an integral part of the mucosal immune system and play a key role in the regulation of host defense and immune tolerance to a multitude of enteric antigens [38]. Mice were orally immunized to characterize the population of dendritic cells in gut-associated lymphoid tissue (GALT) after an oral vaccine. At 12 h and 7 days after the last immunization (Figure 3A), antigen-presenting cells (APC) were immunophenotyped by flow cytometry. The oral vaccine based on *rL. lactis* SIP significantly increased the number of DCs MHC II^high^ CD103 + CD11c+ CD11b+ and MHC II^high^ CD103^‒^ CD11b+ in the GALT in comparison with the PBS and *L. lactis* control groups (* *p* < 0.05) (Figure 3B,D). 

In contrast, no significant increase was observed in the subtype DCs of mice immunized with *L. lactis* in comparison to the PBS control group (ns, *p* > 0.05). However, the percentages of DCs were markedly decreased after administration for 7 days, and no significant differences were detected between mice treated with *rL. lactis*-SIP, *L. lactis,* and PBS (Figure 3C,E). The CD11b subset of CD103+ cells and the CD103−CD11b+ DC subset is related to Th1 and Th17 cell differentiation in response to TLR ligands [39], suggesting that *rL. lactis*-SIP enhances antigen uptake into APCs in the gastrointestinal mucosa for at least 12 h.

### 3.5. Immunization with rL. lactis-SIP Induced T cell Activation and Decreased Regulatory T cells

To evaluate the cellular immune response induced by vaccination with *rL. lactis*-SIP, C57BL/6 mice were immunized with *rL. lactis*-SIP, *L. lactis*, or PBS. Single-cell suspensions were obtained from the spleens of immunized mice and were treated (Figure 4A). As shown in Figure 4A–D, we observed high percentages of CD45+ CD8+ CD69+ and CD45+ CD4+ CD69+ in spleens of mice immunized with *rL. lactis*-SIP (** p < 0.01; *** p < 0.001).). On the contrary, mice immunized with *L. lactis* and PBS showed reduced activation of T cells in their spleens (ns, *p* > 0.05). 

Oral tolerance is a state of systemic unresponsiveness that is the default response to food antigens in the gastrointestinal tract [39]. Antigens can be acquired directly by intestinal phagocytes or pass through enterocytes or goblet cell-associated passages prior to capture by DCs in the lamina propria [40]. In this context, the role of T cells in mediating oral tolerance is induced by antigen-specific CD4+ CD25+ Foxp3+ Tregs (one of the great mediators of tolerance). Our vaccine generated a decrease in Tregs in comparison to the PBS group (Figure 4E–G), which correlates with greater activation of the subset of CD69+ T cells. Furthermore, to support the activation of T cells, we evaluated the gene expression of related cytotoxic genes, STAT3 and STAT5. In this context, STAT3 competes with STAT5 for genome-wide DNA binding, and sustained STAT3 activation may displace STAT5, thereby inhibiting the expression of cytotoxic genes [41]. As shown in Figure 4H,I, there were significant increases in STAT5 in response to *rL. lactis*-SIP and *L. lactis* as in comparison with PBS. Low levels of STAT3 were also observed in cells obtained from the *rL. lactis*-SIP and *L. lactis* groups in comparison with PBS-immunized mice. Additionally, the expression levels of Foxp3 were decreased in *rL. lactis*-SIP-immunized mice in comparison with *L. lactis‒-* and PBS‒immunized mice. Collectively, the activated T cell phenotype observed in splenocytes of *rL. lactis*-SIP and L. *lactis*-immunized mice were compatible with the cellular immune response induced by SIP (TLR agonist). 

### 3.6. Adoptive Humoral and T Cell Transfer from rL. lactis-SIP- Immunized Mice Decreased GBS Colonization

We next evaluated whether T cells in *rL. lactis*-SIP-immunized mice mediated protection against GBS colonization. To this end, splenocytes were obtained from immunized mice and treated in vitro with 5 µg/mL of purified rSIP from *E. coli* to induce activation of Ag-specific T cells (Figure 5A). At day 4 of culture, CD45+ CD25+ cells were measured as a parameter of T cell activation. Significant increases of CD45+ CD25+ T cells in response to stimulation with rSIP could only be detected in cells derived from *rL. lactis*-SIP-immunized mice in comparison with unstimulated splenocytes (data not shown). 

To evaluate whether SIP-specific CD4+ and CD8+ cells have the capacity to confer protection against GBS colonization, purified CD4^+^ and CD8^+^ T cells were isolated from *rL. lactis*-SIP and PBS-immunized mice and transferred to unimmunized C57BL/6 mice (Figure 5A). A single-cell suspension of CD4^+^, CD8^+^, and a 1:1 mixture of these were transferred to naïve mice intravenously. At 24 h after T cell transfer, recipient mice were colonized with GBS. Serum from these mice was obtained and transferred to naïve mice in the same way. This was performed with the aim of determining whether the humoral response could, diminish the colonization of GBS by itself. 

As shown in Figure 5B, mice that received T cells and serum from *rL. lactis-*SIP-immunized mice showed a decrease in GBS vaginal colonization in comparison with mice that received T cells and serum from PBS-immunized mice (** *p* < 0.01). After 3 weeks of colonization by GBS, recipient mice that received CD4 T cells from *rL. lactis*-SIP-immunized mice and colonized with GBS had a high antibody level against SIP in comparison with control animals (**** *p* < 0.0001) (Figure 5C). Additionally, we evaluated whether antibodies generated by recipient mice that received CD4+ T cells from *rL. lactis*-SIP-immunized mice can generate opsonophagocytosis of GBS in vitro according to the OPA. 

This in vitro assay correlates with the effective efficacy of a vaccine for GBS [42]. There was an increase in the kill percentage of GBS in sera obtained from *rL. lactis*-SIP-immunized mice and mice that received CD4+ T cells from *rL. lactis*-SIP-immunized mice and colonized with GBS. These data suggest that passive transfer of humoral and cellular immunity from mice that were orally immunized with *rL. lactis*-SIP into naïve animals can significantly reduce GBS colonization according to vaginal colonization in vivo and the OPA assay in vitro.

## 4. Discussion

GBS is a common colonizer of the female genital tract during pregnancy. It has been associated with severe neonatal infections, and there have been few changes in mortality rates since 1990 [5]. Maternal colonization is a necessary condition for ascending fetal infection, stillbirth, and early onset of the disease in neonates, constituting a risk factor for late-onset disease [43,44]. Currently, there is no licensed vaccine for GBS that can prevent bacterial infection [45,46]. GBS colonizes organisms through mucosal surfaces, and thus an oral vaccination could be an excellent approach to inducing a protective systemic and mucosal immune response [47]. In this context, many studies have highlighted the oral administration of specific probiotic strains (such as *L. lactis*) to enhance both local and systemic immune responses and possibly improve immunization [48]. Moreover, supporting preclinical data have been obtained from clinical trials using HPV-16 (human papillomavirus type 16) vaccines based on *L. lactis* [26]. In this study, we showed that mice immunized with a novel *L. lactis* strain expressing the SIP (*rL. lactis*-SIP) have protective T cell and humoral immunity against GBS. The specificity and efficacy of the vaccine was supported by its capacity to stimulate intestinal DCs and decrease bacterial vaginal colonization. Furthermore, the transfer of specific T cells and serum from *rL. lactis*-SIP-immunized mice led to GBS clearance in recipient animals. This demonstrates that the expression of SIP by *L. lactis* is required to generate bacterial immunity. The mechanism of protection seems to rely on the induction of specific T helper cells that can promote functional immunoglobulins in response to SIP immunizations.

SIP is considered a putative target for the development of a GBS vaccine because it is highly conserved among all GBS serotypes and induces a protective humoral and cellular immune response [13,19]. Moreover, rSIP is an agonist of toll-like receptor 2 (TLR2) and TLR4, which suggests a dual role in the generation of innate and adaptive immune responses against bacterial infection [15]. Previously, we demonstrated that an oral vaccination strategy composed of SIP plus alum adjuvant induces a reduction of GBS vaginal colonization, as well as opsonic antibodies [30]. Oral administration of SIP without an adjuvant generated an increase in IgG and IgA levels, but it was not enough to generate protection [19]. This could be due to the low gastric pH and proteolytic digestion by gastrointestinal enzymes [49]. This suggests that rSIP needs an effective vehicle for administration at the small intestine level to generate protection against GBS. For this purpose, we used *L. lactis* because it is recognized as safe (GRAS), non-pathogenic, and non-colonizing, thus reducing the risk of causing tolerance because of the persistence of mucosal antigen [50]. Additionally, lactic acid bacteria strains have been reported to partially protect against *Streptococcus pyogenes* and *Streptococcus pneumoniae* [51,52]. In this way, the use of *rL. lactis*-SIP has a synergistic effect, because *L. lactis* properties are relative to an unspecific immunity response and SIP has been described as a good immunogen. In this context, the r*L. lactis*-SIP was constructed to achieve direct administration of the protein in the gastrointestinal tract. We demonstrated the ability of *rL. lactis*-SIP to decrease vaginal colonization by GBS in C57BL/6 mice, which correlated with protective cellular and humoral immune responses. This last feature was validated by the protective properties of the vaccine for GBS: increases of IgG, IgA, and serum opsonic antibodies [42]. Therefore, our oral vaccine favors the acceptance of *rL. lactis*-SIP as a possible human immunization vector.

An effective immune response by the mucosa requires the production of serum IgG and secretory IgA (sIgA). At the mucosal level, sIgA acts as the first defense against pathogens through immunological exclusion [53]. This protective function is independent of the cascade of the complement system, which prevents inflammatory damage to the epithelial barrier [54]. The administration of *rL. lactis*-SIP at the gastrointestinal mucosa level generated a specific sIgA response that was detectable in the intestinal lavage and fecal samples from immunized mice. According to their main role in the induction of sIgA, Peyer’s patches (PPs) have an epithelium associated with follicles containing M cells. These are specialized in the capture of antigens [55] and are involved in GBS crossing the intestinal barrier following oral infection in a murine model [56]. Due to its dimensions, *L. lactis* can be easily captured by M cells, which contain an assorted population of lymphocytes, macrophages, and DCs, which in turn organize the IgA response in PPs [57,58]. In mice, CD103+ DCs of the lamina propria could migrate to the mesenteric lymph nodes (MLNs) to initiate the immune response [59]. They have been identified as a subset of DCs characterized by CD11c+ and CD11b+ in the lamina propria that could induce antigen-specific Th1 and Th17 cells, thus contributing to the differentiation of naive B cells into IgA [39]. In this context, oral immunization with *rL. lactis*-SIP increased the percentage of MHC-II+ CD103+ CD11c+ CD11b+ DCs and MHC-II+ CD103^‒^ CD11b+ in the small intestine, along with an increase of sIgA. This suggests that intestinal immunization could induce Th1 and Th17 cell activation and potentially contribute to the differentiation of B cells into IgA plasma cells. 

There are several alternative strategies to prevent or limit maternal GBS colonization in place of the current IAP or proposed vaccine candidates. The alternative strategies explore the growing trend of probiotic agents to limit pathogen overgrowth while promoting healthy native vaginal flora. Multiple studies have documented the inhibitory activity of lactobacilli on GBS growth in vitro. The contribution of *L. lactis* to the growth of GBS has not been totally elucidated. However, GBS is inherently resistant to the antimicrobial activity of nisin, a lantibiotic produced by *L. lactis*, through a nisin-degrading enzyme (SaNSR). In this regard, the use of nisin in our study as an inducer of rSIP secretion by *L. lactis* would not contribute to decreasing GBS vaginal colonization. Due to GBS being found in healthy microbiota at the small intestine level, a plausible question to answer is what the contribution of our vaccine based on *rL. lactis*-SIP is in GBS colonization at the small intestine level. 

Previously, Gupalova et al. [60] developed a live vaccine based on the beta antigen C (Bac) protein of serotypes Ia, Ib, II, V, and IX of GBS. They used the probiotic strain *Enterococcus faecium* L3 for mucosal immunization. An advantage of this method is the increase in the amount of antigen by the probiotic strain due to its ability to multiply at the mucosa level. However, it is necessary to consider the generation of immunological tolerance by constant antigenic stimulation. In contrast, *L. lactis* is a non-invasive and non-pathogenic bacterium with a GRAS state of food grade [48]. Thus, it can be used for the development of live vaccines without the risk of colonization and therefore reduce the risk of generating tolerance. In addition, its peptidoglycan wall is a natural adjuvant [28]. In this context, we observed a significant decrease in GBS colonization in the genitourinary tract of mice immunized with *L. lactis*, which suggests adjuvant characteristics in the decrease of vaginal colonization for GBS. Intrinsic properties of adjuvants have previously been described for the stimulation of a specific immune response against antigens of other pathogens [28]. These results indicate that live oral vaccines using *rL. lactis*-SIP as an expression vector represents a safe, affordable, and easy-to-administer alternative to protect against GBS infection. Serum antibodies contribute significantly to the mucosal defense, mainly because of their opsonophagocytic activity. Furthermore, the specific serum humoral response against SIP could prevent the systemic spread of GBS.

Linked recognition is a process by which a B cell is optimally activated by a helper T cell that responds to the same antigen or a physically associated vaccine [61]. Linked recognition is a well-known rationale for designing vaccines that use capsular polysaccharide antigens, which are poorly antigenic on their own [61]. Our results suggest that SIP can help T cells to produce SIP-specific IgG through linked recognition. Importantly, the anti-SIP antibody secretion at 14 days post-colonization from recipient mice that received CD4+ T cells from *rL. lactis*-SIP-immunized mice are functional according to the OPA assay. This response could be contributed by the capacity of SIP to activate TLR2 and TLR4 [16]. One pending aspect is how CD8+ cells can contribute to decreased colonization. A possible mechanism could involve the induction of cytotoxic T-lymphocytes (CTL) that kill extracellular bacteria, presumably by granulysin and granzymes [62] (Figure 6). On the other hand, because SIP is an agonist of TLR2 and TLR4, it remains unclear whether *rL. lactis* that secretes SIP regulates the intestinal microbiota. It could be involved in the expression of antimicrobial factors such as Reg3γ (regenerating islet-derived protein), thus contributing to the inhibition of pathogenic bacteria [63] (Figure 6). 

## 5. Conclusions

In conclusion, *L. lactis* has been used successfully as a vehicle to stimulate a mucosal immune response against antigens from bacterial and viral pathogens. The use of *L. lactis* as a vector could be considered as a promising vaccine approach against GBS colonization that enhances antigen uptake into APCs in the gastrointestinal mucosa and promotes an efficient humoral response characterized by protective antibodies. Moreover, our *rL. lactis*-SIP strains also induce a cellular immune response that promotes protection when they are transferred to naïve mice. Therefore, given the potential uses of *L. lactis* as a mucosal vaccine vector for a number of pathogens, the vaccine described in our study could be a promising prophylactic approach against GBS colonization in women to prevent invasive neonatal disease and reduce antimicrobial resistance.

## Figures and Tables

**Figure 1 vaccines-08-00146-f001:**
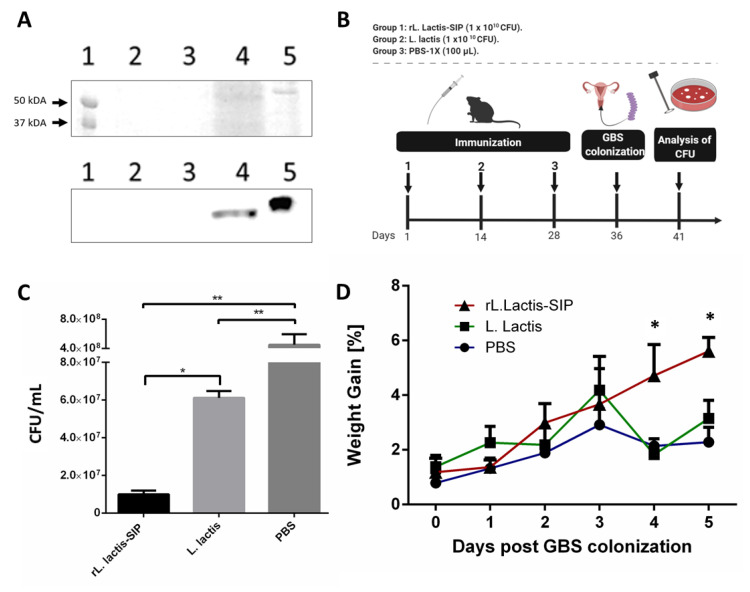
The oral vaccine with recombinant *Lactococcus lactis* strain secreting surface immunogenic protein (SIP) from group B *Streptococcus* (GBS) (*rL. lactis*-SIP) strain diminished GBS vaginal colonization. Groups of C57BL/6 mice received the oral vaccine with 1 × 10^10^ colony forming units (CFU) of *rL. lactis*-SIP, *L. lactis*, or phosphate-buffered saline (PBS) and were infected with 1 × 10^7^ CFU of GBS. (**A**) Analysis of SIP secretion by *rL. lactis*-SIP. After induction with nisin, the rSIP protein produced by *rL. lactis*-SIP was analyzed by SDS-PAGE 12.5% (top) and Western blotting (bottom). Lane 1: Kaleidoscope molecular weight marker (Bio-Rad). Lane 2: Culture supernatant of *L. lactis*. Lane 3: Culture supernatant of *rL. lactis*-SIP without nisin. Lane 4: Culture supernatant of *rL. lactis*-SIP with nisin. Lane 5: Purified rSIP by Ni-NTA (nickel-Nitrilo-Tri-Acetic-acid resin) affinity chromatography column. (**B**) Animal immunization schedule. Representation of the immunization used in C57BL/6 mice experiment. The mice in each (per group) were orally immunized three times with 1 × 10^10^
*rL. lactis*-SIP at intervals of 14 days. The mice in the control groups were immunized with 1 × 10^10^ CFU of *L. lactis* or PBS. (**C**) Vaginal CFU/mL GBS at 5 days post-colonization. Serial dilutions were plated to quantify the CFU of GBS genitourinary tract colonization. Results represent one of two independent experiments with similar results. Data are shown as mean ± SD of four mice in each group. The asterisks indicate a significant difference versus PBS control. Mann–Whitney U-test was used for comparison between treatments (* *p* < 0.05; ** *p* < 0.01). (**D**) Bodyweight gain after GBS vaginal colonization. All mice colonized with GBS maintained a similar weight during the 28 days of immunization (data not shown). Weight loss due to GBS colonization infection for *rL. lactis*-SIP-immunized mice was significantly higher than *L. lactis* and PBS- immunized mice (* *p* < 0.05 by two-way ANOVA).

**Figure 2 vaccines-08-00146-f002:**
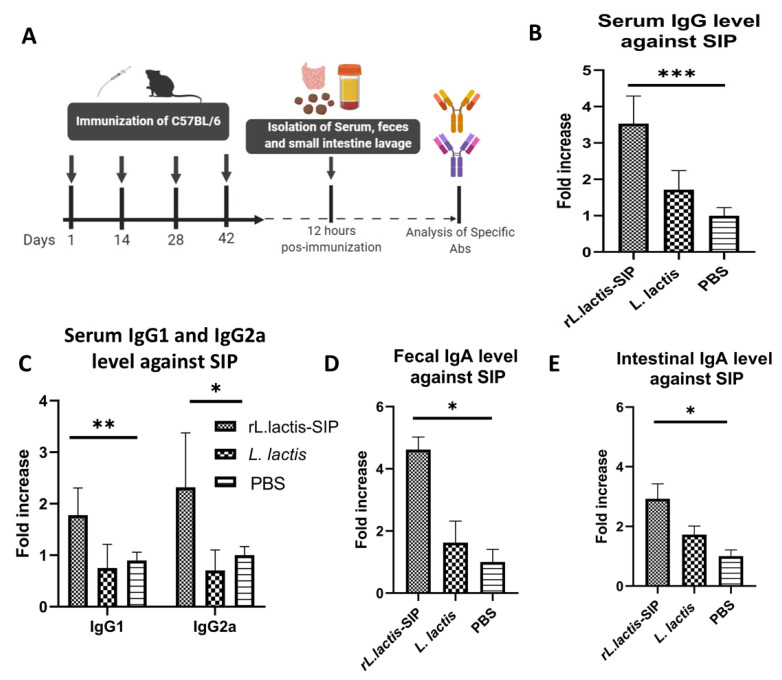
Oral immunization induced with *rL. lactis*-SIP generated systemic and mucosal anti-SIP immunoglobulin G (IgG) and immunoglobulin A (IgA). (**A**) Representation of the immunization used in C57BL/6 mice experiment. The mice (per group) were orally immunized four times with 1 × 10^10^ CFU of *rL. lactis*-SIP at intervals of 14 days. The mice in the control groups were immunized with 1 × 10^10^ CFU of *L. lactis* or PBS. The secretion of specific immunoglobulins against SIP was measured by ELISA from the sera collected from mice immunized without GBS colonization. Serum anti-SIP: (**B**) IgG, (**C**) IgG1 and IgG2a, (**D**) IgA from feces, and (**E**) intestinal IgA production were determined with respect to the PBS-immunized mice (control). Results represent one of two independent experiments with similar results. The bars indicate the mean ± standard deviation of immunoglobulins. *** *p* < 0.0001; ** *p* < 0.001; * *p* < 0.05 by multiple-comparison ANOVA for *rL. lactis*-SIP immunized mice compared with PBS-immunized mice.

**Figure 3 vaccines-08-00146-f003:**
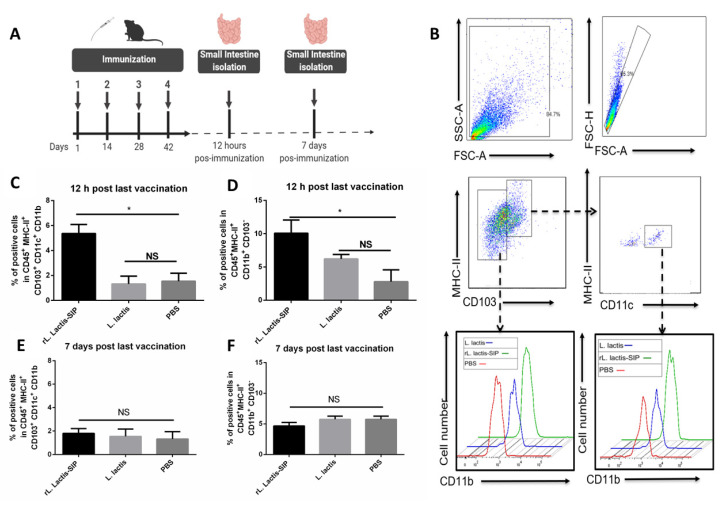
Oral vaccination with *rL. lactis*-SIP enhanced intestinal antigen-presenting cells in gut-associated lymphoid tissue. (**A**) Representation of the immunization used in C57BL/6 mice experiment. The mice (per group) were orally immunized four times with 1 × 10^10^
*rL. lactis*-SIP at intervals of 14 days. (**B**) Representative plots for DCs in the intestine for oral vaccinated mice are shown. Gut-associated lymphoid tissue (GALT) living CD (cluster of differentiation)-45+ MHC (major complex of histocompatibility)-II+ were analyzed for CD103+ and CD103-negative cell expression and consequently the resulting populations were separated for CD11c and CD11b. The mice in the control groups were immunized with 1 × 10^10^ CFU of *L. lactis* or PBS. At 12 h (**C**,**E**) or 7 days (**D**,**F**) after the last immunization, cells were isolated from the entire small intestine and stained with anti CD45- anti-MHC II, anti-CD103+ anti-CD11b+, and anti-CD11c+. Levels of indicated populations of cells expressing various levels of CD11b were analyzed further for CD103 expression. A total of 30,000 events was acquired in all the experiments using a FACSverse cytometer and analyzed with Flowjo software. Results represent one of two independent experiments with similar results. Data are shown as mean ± SD of four mice in each group. Asterisks indicate a significant difference versus PBS control (not significant (ns); * *p* < 0.05).

**Figure 4 vaccines-08-00146-f004:**
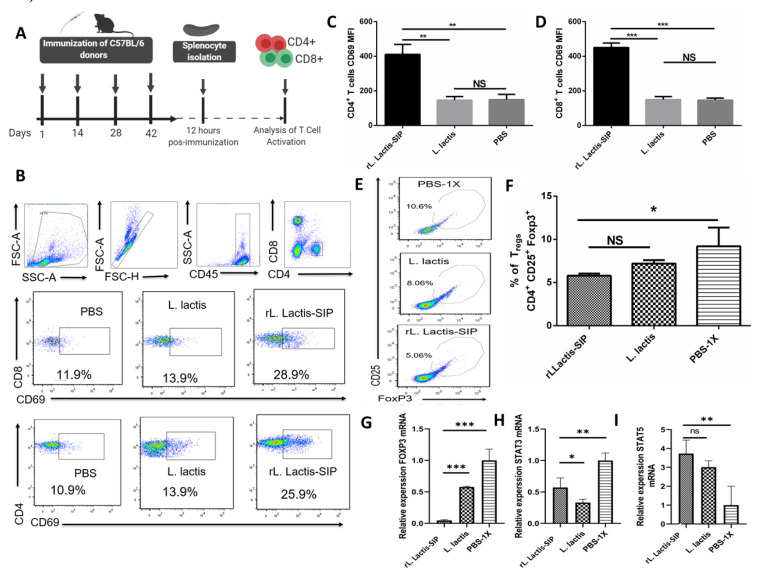
Oral immunization with *rL. lactis*-SIP induced the activation of T cells. (**A**) Representation of the immunization used in the C57BL/6 mice experiment. The mice (four per group) were orally immunized four times with 1 × 10^10^
*rL. lactis*-SIP at intervals of 14 days. (**B**) CD69 expression on the surface of T cells after stimulation with rSIP. At 24 h after the last immunization with either *rL. lactis*-SIP, *L. lactis*, or PBS, spleen cells were collected to evaluate the specific T cell response. Graphs show mean fluorescent index (MFI) of CD69 expression by (**C**) CD4+ and (**D**) CD8+ T cells derived from *rL. lactis*-SIP, *L. lactis*, and PBS-immunized mice. Spleen cells were stained with anti-CD8, anti-CD4, and anti-CD69 antibodies (Abs) and analyzed by flow cytometry. (**E**) Representative flow cytometry plots gated on live CD4 T cells quantify the percentage of CD4+ CD25+ Foxp3+ (forkhead box P3) Tregs obtained from splenocytes. Flow cytometry detection of (**F**) CD4+ CD25+ FoxP3 expression by T cells derived from *rL. lactis*-SIP, *L. lactis*-, and PBS-immunized mice. Spleen cells were stained with anti-CD4, anti-CD25, and anti-Foxp3 Abs and analyzed by flow cytometry. A total of 30,000 events was acquired in all the experiments using a FACSverse cytometer and analyzed with the FlowJo software. Results represent one of two independent experiments with similar results. Data are shown as mean ± SD of four mice in each group. Asterisks indicate a significant difference versus PBS-immunized mice (not significant (ns); * *p* < 0.05; ** *p* < 0.01; *** p < 0.001). Activation of splenocytes obtained from oral-immunization mice was determined by gene profiles of (**G**) Foxp3, (**H**) Signal transducer and activator of transcription (STAT)-3, and (**I**) STAT-5 from total RNA splenocytes. Splenocytes were stimulated ex vivo for 4 days with purified rSIP. Data are expressed as the increase compared to PBS-immunized mice in the expression of the number of Foxp3, STAT-3, and STAT-5 gene copies per β-actin gene copies. Data are shown as mean ± SD of four mice in each group. One of two independent experiments with similar results is shown. Asterisks indicate a difference between groups (not significant (ns); * *p* < 0.05; ** *p* < 0.01; *** *p* < 0.001).

**Figure 5 vaccines-08-00146-f005:**
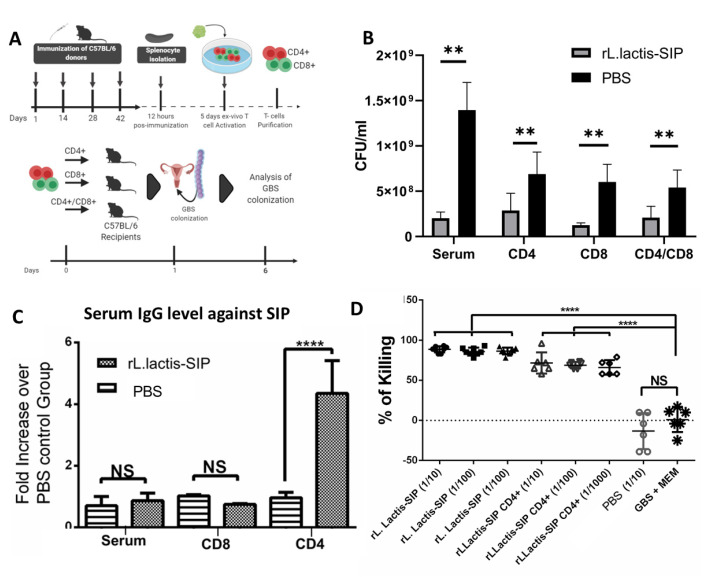
Serum and T cells from *rL. lactis*-SIP promoted a protective immunity to GBS vaginal colonization. (**A**) C57BL/6 mice were orally immunized with *rL. lactis* SIP and PBS and boosted 14, 28, and 42 days later. Then, 12 h after the last immunization, mice were sacrificed, and splenocytes were collected for in vitro culture. Cells were stimulated with 1 µg/mL of purified rSIP and cultured for 4 days. Total cells were pooled, and T cells were purified with Mojosort isolation kits. A total of 1 × 10^6^ purified CD4+, CD8+, or a 1:1 mixture of CD4+ and CD8+ T cells was intravenously injected into naïve mice, and 24 h later, mice were vaginally colonized with 1 × 10^7^ CFU of GBS. Sera from all groups were used to transfer into immunized naïve mice. (**B**) Reduction of GBS vaginal colonization for CD4+, CD8+-, or CD4+/CD8+, and serum from-transferred mice. Data are shown as mean ± SD of eight mice in each group. Asterisks indicate a difference between groups (** *p* < 0.001). (**C**) The secretion of IgG against SIP was measured by ELISA from the sera collected from mice that received T cell and serum transfers and were infected after 4 weeks post-colonization. One of two independent experiments with similar results is shown. Data are shown as mean ± SD of four mice in each group. Asterisks indicate a difference between groups (not significant (ns); **** *p* < 0.0001). (**D**) After 4 weeks, the sera of the recipient animals that received the transfer of CD4 + *rL. lactis*-SIP T cells were tested by opsonophagocytic assay (OPA). OPA was used to test the opsonic killing of GBS from antibodies secreted by the *rL. lactis*-SIP vaccine and CD4+ T cell-transfer mice. The points show the OPA of serum in the presence of GBS, rabbit complement C′, and human promyelocytic leukemia cells (HL-60 cells). The OPA assay was performed with a ratio of effector cells to GBS cells of 90:1. The percentage of killing for the sera and the corresponding dilution used in the OPA are indicated. The control reactions contained dilutions that lacked antibody or complement or contained normal rabbit serum resulted in GBS growth. **** *p* < 0.0001 by multiple-test ANOVA for serum dilutions of 1:10, 1:100, and 1:1000 compared with normal GBS culture in the presence of minimum essential medium (MEM). Data are shown as mean ± SD of eight mice in each group. Asterisks indicate a difference between groups (not significant (ns); **** *p* < 0.0001).

**Figure 6 vaccines-08-00146-f006:**
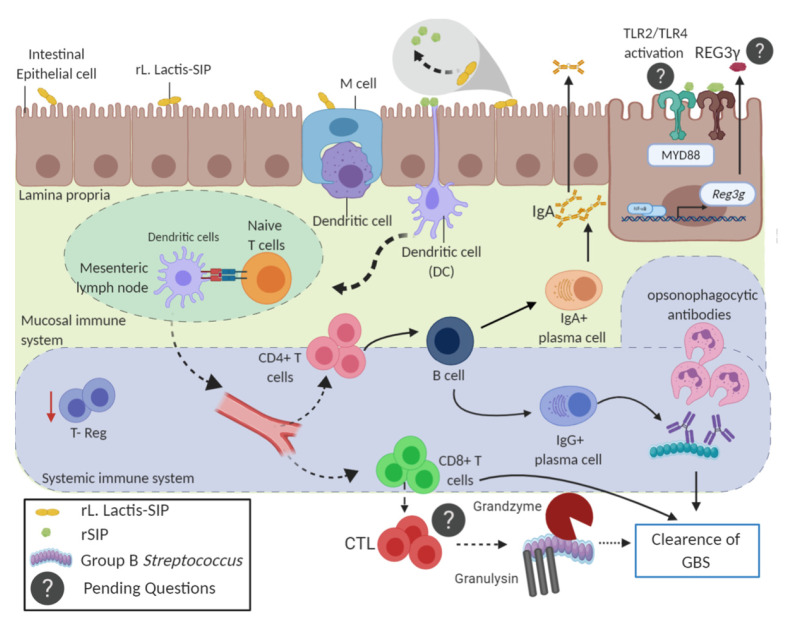
Scheme of the possible humoral and cellular mechanism involved in decreasing GBS colonization by *rL. lactis*-SIP oral vaccination. The function of the oral *rL. lactis*-SIP on stimulation of intestinal dendritic cells that can induce a protective cellular and humoral immunity against group B *Streptococcus*. Following the uptake of SIP from *L. lactis*, intestinal dendritic cells migrate to lymphoid tissue to prime naive CD4+ T cells and CD8+ T cells. Then, T helper cells stimulate IgG and IgA antibody production by B cells. SIP-specific antibodies can be neutralizing and prevent GBS infection. The contribution of CD8 T cells in reducing vaginal colonization remains unclear, as does the way in which SIP could modulate the innate immune response in the intestinal microbiota.

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
