# Peer review of "Mucosal Vaccination with Lactococcus lactis-Secreting Surface Immunological Protein Induces Humoral and Cellular Immune Protection against Group B Streptococcus in a Murine Model"

_vaccines, 2020, doi:10.3390/vaccines8020146_

Round 1
Reviewer 1 Report
The manuscript submitted by Diaz-Dinamarca et al., entitled as “Mucosal Vaccination with Lactococcus lactis secreting Surface Immunological Protein induces Humoral and Cellular Immune Protection against Group B Streptococcus in a murine model.” Attempts to develop an oral vaccine and suggest it to be effective against GBS vaginal colonization in mice model. Further, authors characterize protection mechanism and conclude that edible vaccine induce immunoglobulin production and T cell activation to confer protection.
Authors had tried to do so many different things, but failed to show convincingly effectiveness of the vaccine. Manuscript is poorly prepared, with typos everywhere. Methods of experiments are poorly defined, data is not properly presented, and conclusions are more of speculative in nature. However, manuscript has merits and therefore, this reviewer encourages authors to thoroughly revise and resubmit their manuscript. They need to tone-down their conclusions.
Comments:
- Line 27. colonization of GBS…(“of” is missing)
- Introduction section, authors need to explain why hexavalent conjugate vaccine and protein-based vaccine (line 55) may not be successful? Why edible vaccines are only alternative?
- Line 70. Provide scientific name of tilapia model.
- Line 79. Provide reference for GRAS categorization of L. lactis.
- Methods section 2.3: provide details of primers used for cloning. Details of Genebank accession number of sequence for GBS and L. lactis.
- Methods section 2.2 and 2.5, source of mice are different. Also, authors indicate use of mice at 6-8 weeks of age. For experiments where body weight was measured, Fig. 1E, authors need to indicate that all mice used were of same age and body weight. Mice body weight gain depends on many factors, including age. Therefore, results in Fig. 1E are questionable.
- Line 142: “Institutional Ethic Committee” or CICUAL (line 105), choose one source only.
- Methods section 2.5, Line 143- 147. Why authors chose two different vaccination regimen? Explain rationale.
- Line 157-158: Details of HL-60 cells culture are not provided.
- Reference style in line 159, 167, and 179 is different (why year is shown).
- At many places, CFU had been shown as UFC.
- PBS-1 or PBS-1x, does not make sense. Simply expressing PBS is enough.
- For T cell stimulation, what culture conditions were used? Provide details please.
- Fig. 1A, why molecular weight of purified SIP and SIP from culture sup. is different?
- 1D and 1E: How many mice/group? How and what time of day temperature was measured? A difference of 0.1-0.3 degree C cannot be considered significant? How is this tiny difference physiologically relevant? Similarly, as pointed above, the body weight data is not convincing. Age of animals is important.
- 1C. Please explain, why L. lactis (not expressing SIP) is providing protection, as compared to PBS. Mice were challenged with 1E+07 CFU of GBS. In PBS treated animals, where GBS should be multiplying exponentially, only a fraction of CFU count could be detected. It raises two questions: a, the limit and b, the sensitivity of GBS detection. Having said that, the difference in protection mediated by L. lactis with or without SIP is physiologically insignificant in terms of the role of SIP as a vaccine candidate. Authors need to revisit their statistical analysis. Also, figure label should be CFU not UFC/ml.
- Line 317: Ig1 should be IgG1.
- Similar to fig. 1C, in fig. 2B also L. lactis alone can induce Serum IgG against SIP. It raises question on sensitivity of detection method, or the background level is high.
- Line 338: Tregs are not enteric antigens.
- For flow-cytometric data, clear gating strategy, and representative plot diagram should be shown.
- Line 373-375, conclusion that vaccine generated a decrease in antigen-specific Treg population need reconsideration. Antigen-specificity of Tregs is not tested. As mentioned above, L. lactis alone also reduced Treg population. (fig. 4D). Similarly, L. lactis alone had also reduced Foxp3 mRNA, STAT3 mRNA etc. All these results question the specificity of SIP mediated effect on protection, if any?
- Provide details of how T cell activation by rSIP was induced (Fig. 5A). What culture media was used? What supplements? CD45 is a pan leukocyte marker and represent various forms of T cells, including memory T cells. How does CD45+CD25+ cells indicate a sufficient response?
- Also, authors need to test their vaccine candidate for long term protection. After 3 booster doses, authors are measuring response within a week. What happens if mice are challenged after a month?
- 5B: data should be shown as CFU/ml, as in Fig. 1C.
- 5C: Label “IgG” is missing. For CD4, error bar is too high, how data is still significant? Please revisit your statistical analysis.
- 5D is incomprehensible. Simplify labeling, show data as box and whisker plot.
Author Response
Answers to Reviewer 1.
Major comments and suggestions
- Line 27. colonization of GBS…(“of” is missing)
Answer: As requested by the Reviewer, we modified the manuscript to address English writting (Line 27).
- Introduction section, authors need to explain why hexavalent conjugate vaccine and protein-based vaccine (line 55) may not be successful? Why edible vaccines are only alternative?
Answer: As suggested by the reviewer, we have modified the manuscript to clarify this point (line 56 to 58).
- Line 70. Provide scientific name of tilapia model.
Answer: As suggested by the reviewer, we have modified the manuscript to provide scientific name of tilapia model (line 72).
- Line 79. Provide reference for GRAS categorization of L. lactis.
Answer: As suggested by the reviewer, we have modified the manuscript to provide reference for GRAS categorization of L. lactis (line 81).
- Methods section 2.3: provide details of primers used for cloning. Details of Genebank accession number of sequence for GBS and L. lactis.
Answer: As recommended by the Reviewer, we improved the description of the methodology (line 112 and 120).
- Methods section 2.2 and 2.5, source of mice are different. Also, authors indicate use of mice at 6-8 weeks of age. For experiments where body weight was measured, Fig. 1E, authors need to indicate that all mice used were of same age and body weight. Mice body weight gain depends on many factors, including age. Therefore, results in Fig. 1E are questionable.
Answer As suggested by the reviewer, we clarify this point in the text (Line 108, 141-142). The mice used in this study, already came with an immunization scheme of almost 4 weeks, where it is considered as an adult mouse (Line 325-326), now is Fig D
- Line 142: “Institutional Ethic Committee” or CICUAL (line 105), choose one source only.
Answer: As suggested by the reviewer, we have modified the manuscript to clarify this point (line 104-106).
- Methods section 2.5, Line 143- 147. Why authors chose two different vaccination regimen? Explain rationale.
Answer: As requested by the Reviewer, we modified and edited the manuscript to include information relative to this point (Line 149, Ref 58). The objective was to improve the T immune response, so we generated another booster, instead of testing the T response of animals immunized and colonized with the bacteria.
- Line 157-158: Details of HL-60 cells culture are not provided.
Answer: As recommended by the Reviewer, we improved the description of the methodology (Line 159-162).
- Reference style in line 159, 167, and 179 is different (why year is shown).
Answer: As suggested by the reviewer, we clarify this point in the text and delete the year of the reference in the text. Line164
- At many places, CFU had been shown as UFC.
Answer: As suggested by the reviewer, we edit Figure 1C with the correct legend.
- PBS-1 or PBS-1x, does not make sense. Simply expressing PBS is enough.
Answer: As suggested by the reviewer, we edit the text and simplify the PBS expression.
- For T cell stimulation, what culture conditions were used? Provide details please.
Answer: As recommended by the Reviewer, we improved the description of the methodology (Line 240-247).
- 1A, why molecular weight of purified SIP and SIP from culture sup. is different?
Answer: As suggested by the reviewer, we have modified the manuscript to clarify this point (line 286-289).
- 1D and 1E: How many mice/group? How and what time of day temperature was measured? A difference of 0.1-0.3 degree C cannot be considered significant? How is this tiny difference physiologically relevant? Similarly, as pointed above, the body weight data is not convincing. Age of animals is important.
Answer: As suggested by the reviewer, we reviewed the graph that records the temperature of the mice, we observed a tendency of the mice to maintain the temperature in the presence of colonization with GBS. However, we decided to remove the figure from the manuscript, because we agree with reviewer suggestion. On the other hand, regarding the weight of mice infected with GBS, we decided to design a new figure (1D) with % of weight gain for each animal infected with GBS.
- Please explain, why L. lactis (not expressing SIP) is providing protection, as compared to PBS. Mice were challenged with 1E+07 CFU of GBS. In PBS treated animals, where GBS should be multiplying exponentially, only a fraction of CFU count could be detected. It raises two questions: a, the limit and b, the sensitivity of GBS detection. Having said that, the difference in protection mediated by L. lactis with or without SIP is physiologically insignificant in terms of the role of SIP as a vaccine candidate. Authors need to revisit their statistical analysis. Also, figure label should be CFU not UFC/ml.
Answer: As suggested by the reviewer, we discuss this point in the text (Line 299-302). With respect to the fact that Lactis generates a decrease of GBS colonization, this is something new described by probiotic bacterial strain, and this could be due to its adjuvant capacity of its peptidoglycan walls. However, to generate an effective GBS vaccine, it is necessary that the serum has the ability to stimulate an opsonophagocytosis activity against a GBS, here is against SIP protein.
- Line 317: Ig1 should be IgG1.
Answer: As suggested by the reviewer, we clarify this point in the text and delete the year of the reference in the text (line 343).
- Similar to fig. 1C, in fig. 2B also L. lactis alone can induce Serum IgG against SIP. It raises question on sensitivity of detection method, or the background level is high.
Answer: As suggested by the reviewer, we are agree with to high background observed in our ELISA. This may be due to the use of a polyclonal serum against the SIP protein (there is no commercial one available to date).
- Line 338: Tregs are not enteric antigens.
Answer: As suggested by the reviewer, we edit the manuscript and deleted "Tregs" in the paragraph (line 364).
- For flow-cytometric data, clear gating strategy, and representative plot diagram should be shown.
Answer: As suggested by the reviewer, we edit the figures and add gating strategy and representative plot diagrams (Figure 3 and Figure 4).
- Line 373-375, conclusion that vaccine generated a decrease in antigen-specific Treg population need reconsideration. Antigen-specificity of Tregs is not tested. As mentioned above, L. lactis alone also reduced Treg population. (fig. 4D). Similarly, L. lactis alone had also reduced Foxp3 mRNA, STAT3 mRNA etc. All these results question the specificity of SIP mediated effect on protection, if any?
Answer: According to those suggested by the reviewer, we edit the text and upgrade this point (Line 299-302 and 413-414). Regarding the specificity of SIP to induce protection immunity against GBS, that was supported by opsonophagocytic activity (Fig 5D) and T cell activation (Fig 4 C,D). Moreover, we are aware that L. Lactis generates a certain degree of protection against GBS, which could be due to unspecific immunity stimulation by its peptidoglycan wall.
- Provide details of how T cell activation by rSIP was induced (Fig. 5A). What culture media was used? What supplements? CD45 is a pan leukocyte marker and represent various forms of T cells, including memory T cells. How does CD45+CD25+ cells indicate a sufficient response?
Answer: As recommended by the Reviewer, we improved the description of the methodology (Line 240-247).
- Also, authors need to test their vaccine candidate for long term protection. After 3 booster doses, authors are measuring response within a week. What happens if mice are challenged after a month?
Answer: As suggested by the reviewer, we added information regarding the evaluation of functional antibodies from recipient mice that received CD4 cells from animals immunized with rL. lactis-SIP. Where after 4 weeks, the recipient mice were able to generate opsonophagocytosis. This suggests that the vaccine through CD4 + T cells could be generating immunological memory (Line 481 -483) (Figure 5D).
- 5B: data should be shown as CFU/ml, as in Fig. 1C.
Answer: As suggested by the reviewer, we edit the figure and perform respective normality analyzes (Figure 5B).
- 5C: Label “IgG” is missing. For CD4, error bar is too high, how data is still significant? Please revisit your statistical analysis.
Answer: As suggested by the reviewer, we edit the figure and perform respective normality analyzes (Figure 5C).
- 5D is incomprehensible. Simplify labeling, show data as box and whisker plot.
Answer: As suggested, we improved the graph and put the interest groups (Figure 5D).
We would like to thank the Reviewer again for their time and effort in reviewing this work. We hope that the current revised manuscript will be acceptable for publication in Vaccines.

Reviewer 2 Report
Overall this is a sound manuscript. The methods associated with sections 2.3, 2.4, and 2.11 I can not comment on.
My only concern is associated with Statistical Analyses, from an experimental design only using a two-tailed ANOVA tells the reader very little. Not familiar with GraphPad Prism 5.0--section 2.13 does not provide enough information to assess the statistical model accurately.
Author Response
Answers to Reviewer 2.
Comment and suggestion
- My only concern is associated with Statistical Analyses, from an experimental design only using a two-tailed ANOVA tells the reader very little. Not familiar with GraphPad Prism 5.0--section 2.13 does not provide enough information to assess the statistical model accurately.
Answer: As recommended by the Reviewer, we improved the description of the methodology is associated with Statistical Analyses (Line 217- 276).
We would like to thank the Reviewer again for their time and effort in reviewing this work. We hope that the current revised manuscript will be acceptable for publication in Vaccines.
Round 2
Reviewer 1 Report
Thanks to authors for diligently revising manuscript. Manuscript has improved a lot. Below are my comments which may be helpful to improve the manuscript further.
- Line 57, Description of multivalent vaccine is not helpful. Statement “the same way…as..the protein-based vaccine..” does not aid in our understanding of need for a better vaccine. Authors may like to comment on issues related to antigenicity of protein-based vaccine.
- Section 2.3. “sip” is gene name, should be in italics.
- Line 141, referencing style need attention.
- Section 2.6. Did authors confirm conversion % of HL-60 cells to neutrophil? How pure was the population? Also, ATCC recommend culturing cells in IMDM media, rather than RPMI 1640.
- Line 241, reference is missing.
- This reviewer is not convinced with the claim that rSIP has an effect over L. lactis induced protection (Fig. 1C and 1D). While PBS treated control mice had 2E+08 CFU/ml, both rSIP and L. lactis have 1E+05 CFU. A difference of fraction of 1E+05 is not physiologically relevant and thus not convincing. Authors need to discuss it. There is a big variation in weight gain Fig. 1D, between day 3 and 4, in L. lactis treated mice. How authors will explain it, especially since at day 5, L. lactis had reduced CFU count, but weight gain % on day 5 is close to PBS treated animals. Overall, this data is not very convincing.
- Data in Fig. 3B. the third panel should be of CD103 vs CD11b and CD11c. Figure legend should mention that for which treatment group representative plots are shown. Also, panels should show % population, as done for Fig 4B.
- 4B, panel H & I need attention. For panel C & D, a comparison of CD69 gMFI would have been a better tool. As CD69 expression is dynamic in nature.
- Throughout manuscript "L. lactis" should not be written as "L. Lactis". species name should not be capitalize.
- Fig. 5 looks nice.
Author Response
Answers to Reviewer 1.
Major comments and suggestions
- Reviewer: Line 57, Description of multivalent vaccine is not helpful. Statement "the same way…as..the protein-based vaccine.." does not aid in our understanding of need for a better vaccine. Authors may like to comment on issues related to antigenicity of protein-based vaccine.
Answer: As suggested by the reviewer, we have modified the manuscript to clarify this point (line 57 to 61).
- Reviewer: Section 2.3. “sip” is gene name, should be in italics
Answer: As suggested by the reviewer, we have modified the manuscript and use sip gen name in italics
- Reviewer: Line 141, referencing style need attention
Answer: As suggested by the reviewer, we have modified the manuscript (Line 144)
- Reviewer: Section 2.6. Did authors confirm conversion % of HL-60 cells to neutrophil? How pure was the population? Also, ATCC recommend culturing cells in IMDM media, rather than RPMI 1640.
Answer: As recommended by the Reviewer, we improved the description of the methodology (line 165 and 167). Finally, in the culturing cells both IMDM and RPMI media are used by HL-60 cells. Our methodology was based in reference 28
- Reviewer: Line 241, reference is missing.
Answer: As recommended by the Reviewer, we have modified the manuscript (ref 59)
- Reviewer: This reviewer is not convinced with the claim that rSIP has an effect over L. lactis induced protection (Fig. 1C and 1D). While PBS treated control mice had 2E+08 CFU/ml, both rSIP and L. lactis have 1E+05 CFU. A difference of fraction of 1E+05 is not physiologically relevant and thus not convincing. Authors need to discuss it. There is a big variation in weight gain Fig. 1D, between day 3 and 4, in L. lactis treated mice. How authors will explain it, especially since at day 5, L. lactis had reduced CFU count, but weight gain % on day 5 is close to PBS treated animals. Overall, this data is not very convincing.
Answer: As suggested by the reviewer, we clarify this point in the text (Line 384-386) and modified the figure 1C.
We would like to remark, L. lactis alone is used as stimulant of the immune response. However, rL. lactis-SIP induce a strong immune response than L. lactis alone. The results described here support our communication.
On the other hand, we agree with you about to less weight gain % observed in L. lactis alone, but are several factors that contribute to that parameter, as the timing of immune response development. We describe before that SIP have a putative adjuvant profile and may contribute to a faster establishment of immune response.
- Reviewer: Data in Fig. 3B. the third panel should be of CD103 vs CD11b and CD11c. Figure legend should mention that for which treatment group representative plots are shown. Also, panels should show % population, as done for Fig 4B
Answer: As requested by the Reviewer, we modified and edited the manuscript to include information relative to this point (Line 384-386).
- Reviewer: 4B, panel H & I need attention. For panel C & D, a comparison of CD69 gMFI would have been a better tool. As CD69 expression is dynamic in nature
Answer: As requested by the Reviewer, we modified and edited the manuscript to include information relative to this point (Line 424-425).
- Reviewer: Throughout manuscript "L. lactis" should not be written as "L. Lactis". species name should not be capitalize Reference style in line 159, 167, and 179 is different (why year is shown).
Answer: As suggested by the reviewer, we edited all the text.
- Reviewer: 5 looks nice.
Answer: We welcome your comment
Round 3
Reviewer 1 Report
Dear Authors, thanks for diligently addressing questions raised in the previous version of the manuscript. The modified version 3 looks nice, but raises concern about data quality.
The revised version of manuscript has some questionable data, as compared to the previous two versions. Please note Fig.1C , Fig 4C and 4D, and compare these panels in manuscript version 1(original) 2 and v3. In the v3., please pay attention to the y-axis in Fig. 1C. In previous versions, both empty L.lactis and L.lactis-SIP were in the range of 01x105. Now, in the v3, empty L.lactis is1x107, while -SIP remain same. Please explain this discrepancy. For Fig 4C and 4D, it seems only labeling had been changed, but data points were not (as compare to the previous versions). Please correct these discrepancies, before taking further consideration on the manuscript.Author Response
Answers to Reviewer 1.
Major comment and suggestion
1. Reviewer: Dear Authors, thanks for diligently addressing questions raised in the previous version of the manuscript. The modified version 3 looks nice, but raises concern about data quality.
The revised version of manuscript has some questionable data, as compared to the previous two versions. Please note Fig.1C , Fig 4C and 4D, and compare these panels in manuscript version 1(original) 2 and v3.
In the v3, please pay attention to the y-axis in Fig. 1C. In previous versions, both empty L.lactis and L.lactis-SIP were in the range of 01x105. Now, in the v3, empty L.lactis is1x107, while -SIP remain same. Please explain this discrepancy.
Answer: In order to respond to the reviewer, in version 1 and 2, both empty L lactis and L lactis-SIP were in the range of 1x105. But that was not true, only look similar because of the scale of the Y-axis. In version 3, we only modified the scale of the Y-axis and now we can see differences between bacterial loads in both empty L lactis and L lactis-SIP. Moreover, we observed a significant difference between these experimental groups.
2. Reviewer: For Fig 4C and 4D, it seems only labeling had been changed, but data points were not (as compare to the previous versions).
Answer: As suggested by the reviewer, we modify figure 4c and 4D. We apologize for the discrepancy.
We hope we have answered satisfactorily to all the reviewer's observations, and we appreciate all your suggestions that allowed us to improve our manuscript.
Round 4
Reviewer 1 Report
Authors have adequately addressed all concerns raised by this reviewer. Manuscript should be carefully proofread for typographical errors (e.g. Line 58). Besides that, manuscript can be accepted for the publication.
Author Response
Answers to Reviewer 1.
Minor comment and suggestion
- Reviewer: Authors have adequately addressed all concerns raised by this reviewer. Manuscript should be carefully proofread for typographical errors (e.g. Line 58). Besides that, manuscript can be accepted for the publication.
Answer: As suggested by the reviewer, we have carefully proofread and edited our manuscript.
We hope we have answered satisfactorily to all the reviewer's observations, and we appreciate all observations that allowed us to improve our manuscript.